# The Role of Environmental Conditions on Marathon Running Performance in Men Competing in Boston Marathon from 1897 to 2018

**DOI:** 10.3390/ijerph16040614

**Published:** 2019-02-20

**Authors:** Pantelis T. Nikolaidis, Stefania Di Gangi, Hamdi Chtourou, Christoph Alexander Rüst, Thomas Rosemann, Beat Knechtle

**Affiliations:** 1Exercise Physiology Laboratory, 18450 Nikaia, Greece; pademil@hotmail.com; 2Institute of Primary Care, University of Zurich, 8006 Zurich, Switzerland; stefania.digangi@usz.ch (S.D.G.); christoph.ruest@bluewin.ch (C.A.R.); thomas.rosemann@usz.ch (T.R.); 3Activité Physique: Sport et Santé, UR18JS01, Observatoire National du Sport, Tunis 2020, Tunisia; h_chtourou@yahoo.fr; 4Institut Supérieur du Sport et de l’éducation physique de Sfax, Université de Sfax, Sfax 3000, Tunisia; 5Medbase St. Gallen am Vadianplatz, St. 9001 Gallen, Switzerland

**Keywords:** endurance, performance, sex, environmental conditions, temperature, wind, wet-dry bulb

## Abstract

This study investigated the effects of weather conditions on male performance during the Boston Marathon from 1897 to 2018. A total of 383,982 observations from 244,642 different finishers were analysed using Generalized Additive Mixed Models. All runners, annual top 100 finishers and annual top ten finishers were considered. Weather conditions, on race day, were: average air temperature (°C), precipitations (mm), wet-bulb globe temperature (WBGT) (°C), wind speed (km/h), wind direction (N, S, W, E) and pressure (hPa). These effects were examined in multi-variable models with spline smooth terms in function of calendar year. Temperature, when increasing by 1 °C, was related to worsened performance for all groups (i.e., by 00:01:53 h:min:sec for all finishers, *p* < 0.001). Wind coming from the West, compared to wind coming from other directions, was the most favourable for performance of all groups of finishers. Increasing precipitations worsened performances of top 100 (estimate 00:00:04 h:min:sec, *p* < 0.001) and top 10 finishers (estimate 00:00:05 h:min:sec, *p* < 0.001). Wind speed, when increasing by 1 km/h, was related to worsened performance for all finishers (estimate 00:00:19 h:min:sec, *p* < 0.001), but not for top 100 group, where performances were 00:00:09 h:min:sec faster, *p* < 0.001. Pressure and WBGT were examined in uni-variable models: overall, performances worsened as pressure and WBGT increased. Our findings contributed to the knowledge about the effect of weather conditions on performance level in male marathon runners.

## 1. Introduction

Environmental conditions seem to have an important effect on marathon running performance. Temperature, humidity and wind could influence the thermoregulatory anticipation to the increased heat gained during a marathon race [1]. An ambient temperature higher than 35 °C and a humidity higher than 60% would be detrimental for thermoregulation independent of heat acclimation and optimal nutrition [2]. Also, a high temperature and a high humidity might increase the risk for hyponatraemia, [3] rhabdomyolysis [4] and the percentage of non-finishers [5]. In addition to the abovementioned health implications of weather conditions during a marathon race, temperature, humidity and wind have been identified as factors of performance in this sport [6,7,8] considering their role on heat dissipation. The ranges of temperature 4.4–15.0 °C [7] or 10–12 °C [8] have been recommended as the most advantageous for performance. However, it should be highlighted that the risk for health might increase in temperature higher than 5.5 °C [7] and the recommended ranges might be lower for faster runners [8]. The role of weather conditions has been studied in elite finishers [9] of the Boston Marathon. For instance, it has been shown that their best performances were associated with a temperature lower than ~8 °C and 100% sky cover, whereas their worst performances with temperature higher than ~8 °C and sky cover less than 50% [9]. Furthermore, an analysis of elite runners at the Beijing International marathon reported a very large correlation (r = 0.89) between race time and temperature, i.e., the hotter the temperature the slower the race time [10]. As it has been suggested by previous research, the effect of the temperature on race time might be mediated by the performance level of runners [8]. In addition, light runners might have lower metabolic heat production due to their low body mass and advantageous high surface area to body mass ratio [11]. In this context, the relationship between environmental conditions and marathon race time would offer a framework to study the thermoregulatory ability of humans during endurance exercise.

The Boston Marathon has the longest tradition in marathon running since its start in 1897 and several studies have investigated the influence of environmental conditions on race performance [12,13,14,15]. Mainly, these studies investigated only a short period of this race such as 9 years [12,14] or 36 years [13] and concerned elite and sub-elite performances instead of all finishers [13]. Obviously, there is a lack of investigation for this race for the whole period since 1897 and for recreational runners since the number of recreational and age group runners has increased in the last decades in marathon running [16,17]. A very recent study investigated male performance in the Boston Marathon since 1897 also investigating the influence of environmental conditions [15]. However, only temperatures below and above 8 °C and whether there was precipitation or not were investigated while a detailed analysis of different influences such as average air temperature (°C), precipitations (mm), wet-bulb globe temperature (WBGT) (°C), wind speed (km/h), wind direction (N, S, W, E) and pressure (hPa) were missing. Compared to air temperature, WBGT has been less studied in the existed literature on the effect of environment on endurance exercise; however, its use would provide information about environmental heat stress considering aspects such shade and surface’s color [18]. Precipitation might influence race time since the occurrence of rain has been shown to moderately inversely correlated with variation in race time [5]. Moreover, knowledge of the effect of wind speed and direction on race time would enhance our understanding of whether a high speed wind increased (increased the energy cost of running depending on the wind direction) or decreased race time or whether an absence of wind increased the race time possibly due to a decrease in convective heat loss [19]. With regards to atmospheric pressure, it has been suggested that an elevated pressure might increase the inspired oxygen [20], whereas might be related the development of precipitation [21].

Therefore, the aim of the present study was to examine the effect of weather conditions (i.e., average air temperature, precipitations, WBGT, wind speed, wind direction and pressure) on race time of male runners in the Boston Marathon from 1897 to 2018. This marathon race provided a unique model to conduct research on the effect of weather conditions considering their variability among calendar years. We hypothesized to find difference regarding existing literature when investigating the whole period of time (i.e., 1897 to 2018) and including all male annual finishers as well as the annual top 10 and the annual top 100 male finishers.

## 2. Materials and Methods

### 2.1. Ethical Approval

All procedures used in the study were approved by the Institutional Review Board of Kanton St. Gallen, Switzerland, with a waiver of the requirement for informed consent of the participants given the fact that the study involved the analysis of publicly available data (01/06/2010).

### 2.2. Data Sampling and Data Analysis

The Boston Marathon is the world’s oldest annual marathon, where the course is a point-to-point race with its start in Hopkinton (MA, USA) and finishing on Boylston Street in Boston, MA. (www.baa.org/races/boston-marathon/boston-marathon-history.aspx). To compete in this race, athletes must meet time standards which correspond to age and sex (www.baa.org/races/boston-marathon/participant-information/qualifying.aspx). Data, collected annually from 1897 to 2018 for men were obtained from the official race website (www.baa.org/races/boston-marathon.aspx). Available information from the race records were name and surname of the runners, sex and runners’ nationality, year of competition, and race times.

We cleaned the dataset removing runners with missing or questionable (unreliable) information on race time. Unfortunately, we did not have the complete list of all runners belonging to push rim wheelchair division, started on 1975 for men. We excluded this category eliminating runners with race time shorter than the annual top record. To identify observations from a single runner, we defined an id variable with name, surname, nationality and period of competition, supposing that a single runner could participate at most for 25 years.

Temperature, and speed and direction of wind, seemed to have an influence on race time in the Boston Marathon [9,12,13,14,22], and we therefore merged the data-base with additional information on the weather conditions, during the day of the race, such as average air temperature (°C), total amount of precipitations (mm), WBGT (°C), wind speed (km/h), wind direction (N, S, W, E) and barometric pressure (hPa).

Weather data were obtained from two different websites for this period from 1897 to 2018 from www.wunderground.com/history/airport/KBOS/2013/1/15/MonthlyHistory.html for 1930 to 2018 and from https://w2.weather.gov/climate/local_data.php?wfo=box for 1897 to 1929. WBGT was calculated with https://www.kwangu.com/work/psychrometric.htm using the dry bulb temperature and relative humidity obtained from www.wunderground.com and an altitude of 43 m above sea level.

### 2.3. Statistical Analysis

Descriptive statistics were presented as means ± standard deviations for continuous variables and as number N (%) for categorical variables. Performance, or race time, was recorded in the format “hours:minutes:seconds”. Different analyses were performed for the following groups of finishers: all runners, annual top 100 and annual ten finishers. Average performances, by sex and weather conditions, were reported for the following groups: temperature of 0–7, 8–15, 16–23, 24–30 °C, precipitations equal 0 or >0 mm, WBGT of 0–6, 7–10, 11–15, 16–20 °C, wind speed of 0–15, 16–17, 18–38 km/h, and pressure < 1015 or ≥1015 hPa. Wind direction was defined as: East (E, ENE, ESE), West (W, WNW, WSW), North (N, NE, NNE, NNW, NW, NWW), South (S, SE, SSE, SSW, SW, SWS). T-tests or ANOVA tests, with post-hoc Dunn’s test for pairwise multi-comparison of means, where appropriate, were performed, but p-values were not shown in tables.

Data visualization was used to identify the uni-variable relationship between performance and each weather variable. Moreover, association between weather conditions was preliminary investigated through the correlation matrix, which showed the pairwise correlation among all the variables. As a result, the effects of calendar year on race time, together with the effects of weather conditions, were examined through multi-variable statistical models. Results were presented as estimates (standard errors). The acceptable type I error was set at *p* < 0.05.

Different models were performed, one for each group of finishers. Calendar year was considered as a discrete value of a continuous variable. Temperature, precipitation, WBGT, wind speed and pressure were considered as continuous variables. Spline regression models were used, with a spline smooth term in function of calendar year and a linear term in function of the other effects. Preliminary data visualization and previous research [15,23] justified the choice of spline regression models for the underlying temporal trend of performance of all groups of finishers. The linear trend, for all weather predictors, was a trade-off between keeping the multi-variable model simple and interpretable and accurately describing the effect of each predictor. To account for repeated measurements within runners, a mixed model, with random effects on intercept for each runner, was performed. In summary, we used Generalized Additive Mixed Models, specified as follows:
Race Time (Y) ~ [Fixed effects (X) = Weather Conditions + S(YEAR, k)+ [Random effects of intercept = Runners]
where S(YEAR, k) denoted a spline, changing with calendar year and with k basis dimension. K was set equal to nine. Observations before 1944 were discarded because of incomplete information about weather conditions. As a first analysis, all weather variables: temperature, WBGT, precipitations, wind direction, wind speed, and pressure were included into the multi-variable model. Subsequently, WBGT and pressure were discarded, in all groups of finishers, due to multi-collinearity, assessed by computing the variance inflation factor (or VIF) score. All predictors with VIF not greater than 10 were retained in the model. According to this criterion, in all finishers group, precipitations variable was also dropped. Instead, in top ten finishers the wind speed variable was discarded. Partial effects, which are the isolated effects of one particular predictor or interaction, were shown graphically. Moreover, the effects of pressure and WBGT on performance, discarded in the multi-variable model, where also assessed, more rigorously, through uni-variable linear mixed models, which corrected for repeated measurements.

All statistical analyses were carried out using statistical package R, R Core Team (2016). R: A language and environment for statistical computing. (R Foundation for Statistical Computing, Vienna, Austria. URL: https://www.R-project.org/). In particular, we used the following R packages: PMCMR for Pairwise post-hoc Test for Multiple Comparisons of Means; ggplot2 for preliminary data visualization; GGally for plotting the correlation matrix and the association between explanatory variables; gamm4 for multi-variable mixed models with random effects on intercept; mgcv for statistical models visualization; lmer for linear mixed models.

## 3. Results

### 3.1. Summary Statistics

Between 1897 and 2018, a total of 383,982 observations from 244,642 different finishers were recorded in the race results. In Table 1, the summary statistics of the average performance, by weather conditions and for all groups of finishers, were reported. Considering the effect of average temperature alone, performances were fastest, on average, in the top 100 (02:34:57 ± 00:18:16 h:min:sec) and top 10 finishers groups (02:23:41 ± 00:12:01 h:min:sec), when the temperature was lower than 8 °C. In the all finishers group, the fastest average performance (03:11:14 ± 00:11:11 h:min:sec) was observed when temperature was in the range 24–30 °C. However, this value was the mean of only 20 observations, the only ones recorded in 1909, where participation was low and the average temperature, 28 °C, was the highest, compared to the temperature observed in other years. Descriptive statistics, gave no hint about the most favourable wind direction in terms of performances. In fact, compared to the other wind directions, better performances were observed: when the wind came from the North in all finishers group (03:32:12 ± 00:40:35 h:min:sec), when the wind came from the South in top 100 group (02:31:28 ± 00:13:32 h:min:sec) and when the wind came from the East (02:18:22 ± 00:09:10 h:min:sec) in top 10 group. These results were shown graphically through box plots (Figure 1, Figure 2 and Figure 3). However, p-values from pairwise multiple comparisons tests between wind direction gave no evidence of the most favourable wind in top 100 and top 10 groups. Performances of all groups, on average, were better when the total amount of precipitations was positive, compared to no precipitations: 03:38:11 ± 00:43:58 h:min:sec in all finishers, 02:34:52 ± 00:17:49 h:min:sec in top 100 group and 02:24:20 ± 00:13:26 h:min:sec in top ten group. When considering WBGT, better performances were observed when the level was 7–10 °C in all finishers (03:37:20 ± 00:39:09 h:min:sec) and 0–6 °C in top 100 group (02:32:40 ± 00:17:19 h:min:sec). In top 10 group, pairwise comparisons between WBGT categories were not significant. Wind speed 16–17 km/h was the most favourable category, in terms of performances: 03:38:27 ± 00:42:28 h:min:sec in all finishers group, 02:30:14 ± 00:13:28 h:min:sec in top 100 group and 02:16:40 ± 00:09:59 h:min:sec in top 10 group. Performances were also better when pressure < 1015 hPa: 03:38:24 ± 00:43:28 h:min:sec in all finishers, 02:31:10 ± 00:15:06 h:min:sec in top 100 group and 02:19:17 ± 00:11:02 h:min:sec in top 10 group. However, in top 10 group, t-test between means of performance, when pressure was below or at least 1015 hPa, was not significant (*p* = 0.348, not shown in tables). In Appendix A, Table A1, race time of all finishers was categorized into equal-spaced groups of half an hour, from 02:00:00 to 06:00:00, except the last group of performances slower than 06:00:00. Then, the frequency distribution of these groups was shown by weather conditions.

### 3.2. Statistical Models

#### 3.2.1. Weather Conditions

Before carrying out multi-variable analysis, association between weather variables was examined and reported in Appendix A, Figure A1. The highest coefficient of correlation, 0.859, was observed between WBGT and average temperature. Then, follow the coefficients of correlation between wind speed and precipitations, 0.321, and between pressure and precipitations, −0.308. Moreover, in Figure A1, density plots of each weather variable were shown to identify skewness, kurtosis and distribution information.

After that, the effects of each weather variable, on performance of all groups of finishers, were shown in Figure 1, Figure 2 and Figure 3. In these figures, three different trend curve hypotheses (linear, quadratic, spline) were considered to model the uni-variable relationship between performance and each weather variable. These plots showed that, for all groups of finishers, the uni-variable relationships between performance and temperature and between performance and wind speed might be better described with spline curves. However, to keep the model simple and immediate to interpret and to prevent the multi-collinearity, the linear trend hypothesis was chosen for each explanatory variable.

In Table 2, the results of the multi-variable generalized additive mixed models, described in the methods section, were shown. When temperature increased, performances of all groups of finishers worsened. In fact, when average temperature increased by 1 °C, performances were slower by: 00:01:53 (00:00:01) h:min:sec, *p* < 0.001 in all finishers; 00:00:37 (00:00:02) h:min:sec, *p* < 0.001 in top 100 finishers and 00:00:38 (00:00:03) h:min:sec, *p* < 0.001 in top 10 finishers. When wind speed increased by 1 km/h, performances of all finishers worsened but performances of top 100 group improved. In fact, the wind speed estimates was positive in all finishers: 00:00:19 (00:00:01) h:min:sec, *p* < 0.001 and negative in top 100 group: −00:00:09 (00:00:01) h:min:sec, *p* < 0.001. When precipitations increased by 1 mm, performances significantly worsened: by 00:00:04 (00:00:01) h:min:sec, *p* < 0.001 in top 100 group and by 00:00:05 (00:00:01) h:min:sec, *p* < 0.001 in top 10 group. Performances with wind coming from the West, were significantly better compared to the other wind directions (*p* < 0.001). In fact, estimates were positive, which meant worse performance compared to the West, the reference wind direction category. The wind direction had the greatest effects, in terms of estimates, in all finishers, compared to the other groups of finishers, when the wind came from the East and from the South. In fact, when the wind direction was the East, performances, compared to performances when wind direction was the West, were slower by: 00:11:18 (00:00:11) h:min:sec, *p* < 0.001, in all finishers, against 00:03:09 (00:00:27) h:min:sec, *p* < 0.001, in top 100 group and 00:03:54 (00:00:38) h:min:sec, *p* < 0.001 in top 10 group. Analogously, when the wind direction was the South, performances, compared to performances when wind direction was the West, were slower by 00:09:41 (00:00:10) h:min:sec, *p* < 0.001 in all finishers, against 00:01:13 (00:00:33) h:min:sec, *p* < 0.05 in top 10 group. In top 100 group, the difference between performances, when wind direction was the South and performances when wind direction was the West, was not significant. When the wind direction was the North compared to the West, the greatest difference was observed in all finishers: 00:05:30 (00:00:11) h:min:sec, *p* < 0.001 against 00:01:51 (00:00:38) h:min:sec, *p* < 0.001 in top 10 group and 00:01:30 (00:00:25) h:min:sec, *p* < 0.001 in top 100 group.

Since the uni-variable relationships, plotted in Figure 1, Figure 2 and Figure 3, did not take into account the repeated measurements within runners, we examined, more rigorously through linear mixed models, the uni-variable relationship between pressure and performance and between WBGT and performance. Results of these models were not shown in tables, but reported as follows. When pressure increased by 1 hPa, performances of all groups significantly worsened by 00:00:09 (00:00:00) h:min:sec, *p* < 0.001 in all finishers, 00:00:09 (00:00:01) h:min:sec, *p* < 0.001 in top 100 groups and 00:00:04 (00:00:02) h:min:sec, *p* < 0.05 in top ten group. Instead, when WBGT increased by 1 °C, performances of all groups of finishers were significantly slower: by 00:01:55 (00:00:01), *p* < 0.001 in all finishers, by 00:00:22 (00:00:03) h:min:sec, *p* < 0.001 in top 100 group and by 00:00:26 (00:00:06) h:min:sec, *p* < 0.001 in top 10 group.

#### 3.2.2. Calendar Year Effect and Weather Conditions

Performances changed significantly over calendar year. In fact, in Table 2, smoothing terms overall had *p* < 0.001 in all groups. The greatest smoothing terms, in absolute value, were observed in all finishers where the trend was overall increasing, even if in some years (i.e., between 1940 and 1950, between 1960 and 1970 and between 2000 and 2010) was decreasing. This meant that performances had overall worsened over time. In top 100 and top 10 groups, instead, the temporal trend of performance was overall decreasing, even if not monotonically, meaning that performance had overall improved over time.

To visualize the most important results of Table 2, in Figure 4, multi-variable effects of temperature and calendar year, on performances of all groups of finishers, were shown through perspective plot views. Therefore, one could observe the year trend marathon and the slowing-down of performance when average temperature increased. In Figure 5, instead, the partial effects of temporal trend, keeping the other predictors constant, were plotted by wind direction for all groups of finishers. Curves were parallel, because in our model specification no interaction between year and weather conditions was considered.

## 4. Discussion

The main findings of the present study were that: (i) average temperature, when increasing by 1 °C, significantly (*p* < 0.001) worsened performance by 00:01:53 h:min:sec for all finishers, 00:00:37 h:min:sec for annual top 100 finishers and 00:00:38 h:min:sec for annual top 10 finishers; (ii) wind coming from the West, compared to wind from other directions, was the most favourable for performance of all groups of finishers. In particular, for all finishers, performances when wind came from the West were: 00:09:41 h:min:sec faster (*p* < 0.001), compared to performances when wind came from the South, 00:11:18 h:min:sec faster (*p* < 0.001), compared to performances when wind came from the East and 00:05:30 h:min:sec faster (*p* < 0.001), compared to performances when wind came from the North; (iii) precipitations, when increasing by 1 mm, worsened performances of top 100 and of top 10 groups by 00:00:04 h:min:sec and 00:00:05 h:min:sec (*p* < 0.001), respectively; (iv) wind speed, when increasing, also worsened performances for all finishers (estimate 00:00:19 h:min:sec, *p* < 0.001), but not for top 100 group, where performances were 00:00:09 h:min:sec faster, *p* < 0.001; (v) pressure, when increasing, had a negative effect on performance in all group of finishers; (vi) WBGT when increasing, had a negative effect on performance of all groups of finishers.

The direct relationship of race times with temperature, i.e., the hotter the temperature, the slower the race time, was in agreement with previous observations in marathon races [5,12]. For instance, it has been previously shown that temperature and performance were correlated through a quadratic model and the optimal temperature for maximal average race speed depended on performance level [12]. Furthermore, the observation that the temperature had larger effect on performance in all finishers than in annual winners confirmed existing literature, where the largest magnitude of correlation between temperature and race time was observed in the slowest marathon runners [5]. This variation by performance level might be partially attributed to a more favourable thermoregulatory profile in elite athletes as result of long-term endurance training [24]. Moreover, it was found that increased WBGT was related to slower race time of all finishers. The similar direction of the effect of WBGT with the one of ambient temperature might be due to the formula that calculated WBGT, which included information on the ambient temperature [18]. Decreased performance due to weather conditions might be explained by the well-known effect of heat on the cardiovascular (e.g., elevated heart rate) and thermoregulatory systems (e.g., elevated core temperature) [3]. Compared to exercise in the cold, exercise in the heat was shown to increase oxygen uptake for a given exercise intensity resulting in increased use of glycogen and lactate production [25]. Moreover, a high temperature might induce fatigue through its effect on central nervous system, and increase dehydration, which in turn, might reduce stroke volume, cardiac output, blood pressure and blood flow to skeletal muscles [3]. Furthermore, it has been shown that heat production during aerobic exercise was directly related to body mass [26] and fast runners were lighter than their slower counterparts [27], which might explain the smaller magnitude of hot temperature for the fastest runners. With regards to the findings about the direction and magnitude of the wind previous observations were confirmed [28]. The race starts in Hopkinton, located to the west-south-west of Boston. The athletes run in the general direction west-south-west and wind coming from west, compared to wind from other directions, was the most favourable for performance of all groups of finishers. Considering the precipitations, an increased amount of rain was associated with slower race time. A similar finding had also been observed in the Stockholm marathon and had been attributed to the negative correlation of precipitation with temperature [5]. Another explanation of the role of precipitations might be that they acted as the same physiological strain as cold with both cooling the human body [29].

The findings of the present study were limited by the specific characteristics of the race in terms of participation [15] and weather conditions of Boston; thus, caution would be needed to generalize the findings to marathon races conducted in other geographical regions. It should be highlighted that the applied multi-variable statistical models considered the effects of calendar years and weather conditions on race time. This statistical approach allowed partitioning out the variation of race time across calendar years, since it has been well documented that the increased number of finishers resulted in overall slower marathon race times [15]. A limitation is also that we used daily but not hourly weather data from the start of the race until the end of the race. This might have had an influence on the outcome of the results. Moreover, only men were considered and consequently, the results should not be applicable to women considering the larger surface area to body mass ratio and slower race time of the latter [5]. Also, it was acknowledged that the classification of marathon runners into performance groups such as top 10 and top 100 referred to different relative numbers of runners (as percentage of the total number) across calendar years. For instance, top 10 was corresponding to a gradually decreased percentage of runners (i.e., became more “elite”) as the number of finishers increased across calendar years. However, these performance groups were selected for the present analysis due to their practical relevance for marathon and their use in the existing literature [30,31]. On the other hand, strength of this study was its novelty as it was the first time to examine several weather conditions combined with various performance levels in the whole sample of men finishers in the Boston Marathon, the world’s oldest annual marathon. Future studies need to confirm these findings for other large city marathons such as the Berlin Marathon or the New York City Marathon. For athletes and coaches, the selection of a marathon race with optimal environmental conditions (e.g., low temperature, low wind, no precipitations) might be important to achieve a fast marathon race time.

## 5. Conclusions

In summary, race times in the Boston Marathon are influenced by temperature, pressure, precipitations, WBGT, wind coming from the West and wind speed. The magnitude and the direction of the influence of the abovementioned weather conditions on race time varied by performance level. Our findings improved the knowledge about the association between weather conditions and their relationship with performance level in male marathon runners. Considering the popularity of the particular marathon and of marathon races in general [32], the results would have important practical applications for runners and practitioners working with them. Although the results could be applied directly only to runners of the Boston Marathon, it was assumed that similar patterns of the relationship between weather conditions and race time would be expected in other marathon races, too.

## Figures and Tables

**Figure 1 ijerph-16-00614-f001:**
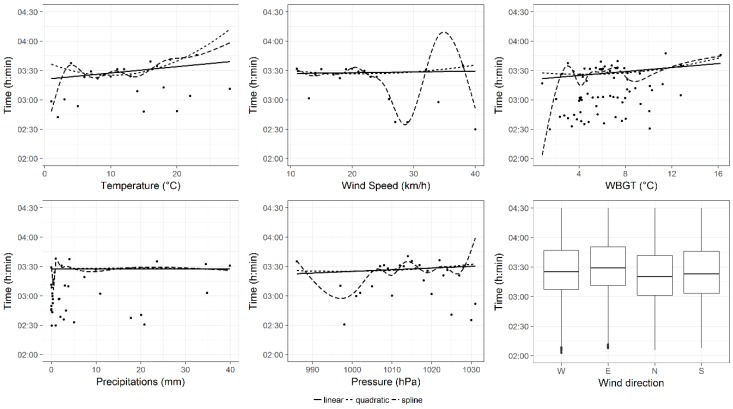
Effects of weather conditions on all finishers’ performance. Curves were the fitted values of uni-variable model smooths with different trend hypotheses (linear, quadratic and spline). Points were observed (mean) values. For wind direction (categorical variable) a box-plot was reported.

**Figure 2 ijerph-16-00614-f002:**
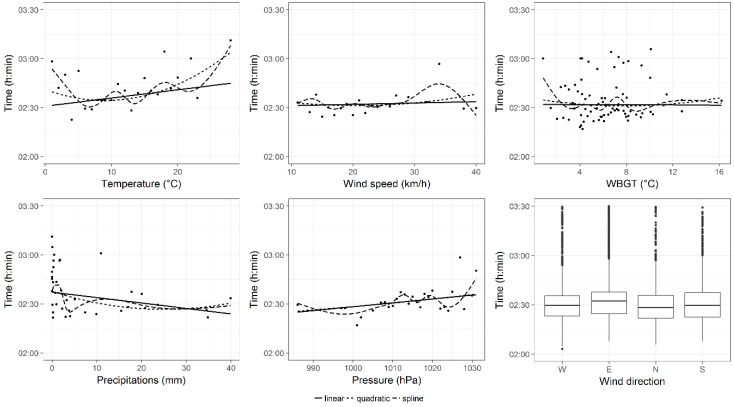
Effects of weather conditions on top 100 finishers’ performance. Curves were the fitted values of uni-variable model smooths with different trend hypotheses (linear, quadratic and spline). Points were observed (mean) values. For wind direction (categorical variable) a box-plot was reported.

**Figure 3 ijerph-16-00614-f003:**
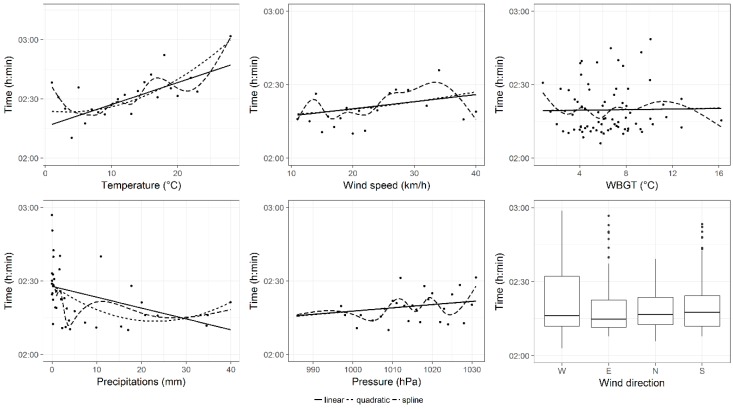
Effects of weather conditions on top 10 finishers’ performance. Curves were the fitted values of uni-variable model smooths with different trend hypotheses (linear, quadratic and spline). Points were observed (mean) values. For wind direction (categorical variable) a box-plot was reported.

**Figure 4 ijerph-16-00614-f004:**
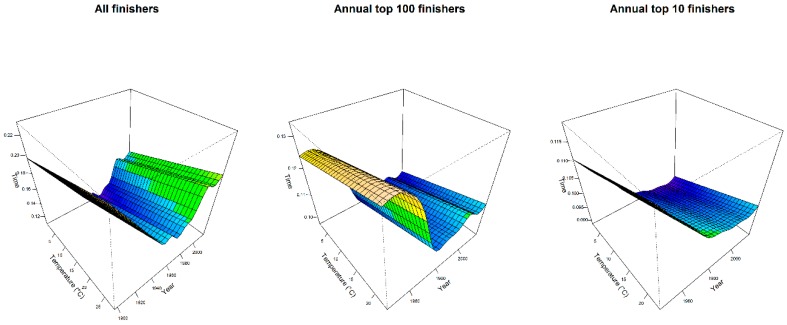
Multi-variable model of performances of all finishers, top 100 and top 10 annual finishers: perspective plot views. Effects of temperature and calendar year. Time was labelled in fraction of a day, i.e., 0.125 = 03:00:00 h:min:sec.

**Figure 5 ijerph-16-00614-f005:**
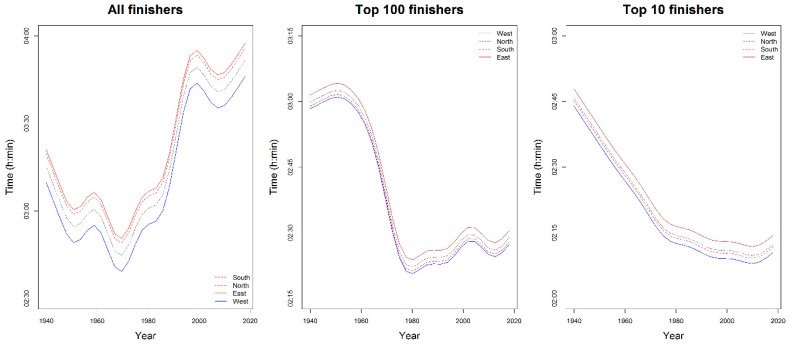
Multi-variable model of performances of all finishers, top 100 and top 10 annual finishers: year trend by wind direction.

**Table 1 ijerph-16-00614-t001:** Race time (Mean and SD) in h:min:sec, by weather conditions, for all groups of finishers.

	All Finishers (*N* = 383,982)			Top 100 (*N* = 7713)			Top 10 (*N* = 1215)		
Temperature (°C)	*N*	Mean	SD	*N*	Mean	SD	*N*	Mean	SD
0–7	116,180	03:41:51	00:44:41	1817	02:34:57	00:18:16	259	02:23:41	00:12:01
8–15	217,649	03:34:37	00:40:18	4851	02:36:49	00:19:20	728	02:26:44	00:14:44
16–23	50,133	04:02:55	00:48:43	1025	02:46:58	00:22:19	218	02:40:55	00:21:15
24–30	20	03:11:14	00:11:11	20	03:11:14	00:11:11	10	03:01:43	00:05:06
**Wind direction**	***N***	**Mean**	**SD**	***N***	**Mean**	**SD**	***N***	**Mean**	**SD**
West	57,966	03:36:23	00:38:54	1103	02:32:25	00:16:14	140	02:22:03	00:14:03
East	171,591	03:45:03	00:43:32	1996	02:35:26	00:18:43	220	02:18:22	00:09:10
North	63,408	03:32:12	00:40:35	1351	02:31:51	00:17:15	160	02:18:41	00:07:59
South	89,567	03:41:01	00:48:13	1813	02:31:28	00:13:32	220	02:20:16	00:10:32
missing	1450			1450			475		
**Precipitations (mm)**	***N***	**Mean**	**SD**	***N***	**Mean**	**SD**	***N***	**Mean**	**SD**
0	215,350	03:42:19	00:43:38	4330	02:39:58	00:20:58	735	02:31:01	00:16:36
>0 mm	168,622	03:38:11	00:43:58	3373	02:34:52	00:17:49	470	02:24:20	00:13:26
missing	10			10			10		
**WBGT (° C)**	***N***	**Mean**	**SD**	***N***	**Mean**	**SD**	***N***	**Mean**	**SD**
0–6	184,772	03:39:27	00:44:57	2969	02:32:40	00:17:19	360	02:19:09	00:09:48
7–10	152,114	03:37:20	00:39:09	2535	02:33:05	00:16:53	290	02:19:33	00:10:48
11–15	33,025	03:49:33	00:47:24	659	02:33:47	00:13:38	80	02:23:14	00:12:24
16–20	12,621	04:15:03	00:52:05	100	02:34:15	00:08:13	10	02:15:26	00:02:06
missing	1450			1450			475		
**Wind speed (km/h)**	***N***	**Mean**	**SD**	***N***	**Mean**	**SD**	***N***	**Mean**	**SD**
0–15	90,938	03:41:32	00:44:16	1536	02:36:26	00:20:06	180	02:21:15	00:11:28
16–17	104,511	03:38:27	00:42:28	1534	02:30:14	00:13:28	160	02:16:40	00:09:59
18–38	187,066	03:41:29	00:44:17	3176	02:33:03	00:16:32	400	02:20:50	00:10:46
missing	1467			1467			475		
**Pressure (hPa)**	***N***	**Mean**	**SD**	***N***	**Mean**	**SD**	***N***	**Mean**	**SD**
<1015	167,243	03:38:24	00:43:28	2741	02:31:10	00:15:06	330	02:19:17	00:11:02
≥1015	215,289	03:42:25	00:44:01	3522	02:34:23	00:17:41	410	02:20:02	00:10:04
missing	1450			1450			475		

**Table 2 ijerph-16-00614-t002:** Statistical models: estimates (std errors) are reported in h:min:sec. For p-value see Note. For smoothing terms, denoted with S(YEAR)n, *n* = 1–9, a global t-test was reported. For the categorical predictor, the reference category (ref.) was reported.

	All Finishers	Top 100	Top 10
(Intercept)	03:09:56 ***	02:28:34 ***	02:11:26 ***
	(00:00:22)	(00:00:50)	(00:00:50)
**Temperature (°C)**	00:01:53 ***	00:00:37 ***	00:00:38 ***
	(00:00:01)	(00:00:02)	(00:00:03)
**Precipitations (mm)**		00:00:04 ***	00:00:05 ***
		(00:00:01)	(00:00:01)
**Wind speed (km/h)**	00:00:19 ***	−00:00:09 ***	
	(00:00:01)	(00:00:01)	
**Wind direction (ref. West)**			
Wind direction = E	00:11:18 ***	00:03:09 ***	00:03:54 ***
	(00:00:11)	(00:00:27)	(00:00:38)
Wind direction = N	00:05:30 ***	00:01:30 ***	00:01:51 **
	(00:00:11)	(00:00:25)	(00:00:38)
Wind direction = S	00:09:41 ***	00:00:35	00:01:13 *
	(00:00:10)	(00:00:22)	(00:00:33)
**Smoothing terms**	*p* < 0.001 overall	*p* < 0.001 overall	*p* < 0.001 overall
S(YEAR)1	−00:02:45 ***	−00:09:39 ***	00:00:24 ***
	(00:04:50)	(00:01:19)	(00:01:37)
S(YEAR)2	00:24:34 ***	−00:32:16 ***	00:08:14 ***
	(00:14:19)	(00:04:30)	(00:04:37)
S(YEAR)3	00:30:56 ***	00:08:31 ***	00:00:25 ***
	(00:04:21)	(00:00:51)	(00:01:21)
S(YEAR)4	00:48:38 ***	−00:01:40 ***	00:07:55 **
	(00:10:19)	(00:03:23)	(00:03:26)
S(YEAR)5	−00:35:50 ***	00:02:35	00:02:50 *
	(00:04:25)	(00:00:58)	(00:01:23)
S(YEAR)6	−00:44:38 ***	00:00:16 ***	−00:08:02 ***
	(00:08:33)	(00:03:17)	(00:03:10)
S(YEAR)7	−00:05:23 ***	−00:06:37 ***	00:00:02 ***
	(00:05:44)	(00:01:05)	(00:01:19)
S(YEAR)8	−01:55:53 ***	00:14:43 ***	−00:24:31 ***
	(00:27:54)	(00:07:53)	(00:07:09)
S(YEAR)9	−00:05:23 ***	00:09:49 ***	−00:04:17 ***
	(00:04:27)	(00:03:04)	(00:03:04)
Observations	**382,532**	**6263**	**740**

Note: * *p* < 0.05; ** *p* < 0.01; *** *p* < 0.001.

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
