# Peer review of "The Role of Environmental Conditions on Marathon Running Performance in Men Competing in Boston Marathon from 1897 to 2018"

_ijerph, 2019, doi:10.3390/ijerph16040614_

Round 1
Reviewer 1 Report
improve discussion of your data, there ara a lot of data in results sections but the discussion is insuficient
Author Response
Reviewer 1
Open Review
(x) I would not like to sign my review report
( ) I would like to sign my review report
English language and style
( ) Extensive editing of English language and style required
(x) Moderate English changes required
( ) English language and style are fine/minor spell check required
( ) I don't feel qualified to judge about the English language and style
Yes | Can be improved | Must be improved | Not applicable | |
Does the introduction provide sufficient background and include all relevant references? | ( ) | (x) | ( ) | ( ) |
Is the research design appropriate? | ( ) | ( ) | (x) | ( ) |
Are the methods adequately described? | ( ) | (x) | ( ) | ( ) |
Are the results clearly presented? | ( ) | (x) | ( ) | ( ) |
Are the conclusions supported by the results? | ( ) | (x) | ( ) | ( ) |
Comments and Suggestion for Authors
Improve discussion of your data, there are a lot of data in results sections but the discussion is insufficient
Answer: We agree with the expert reviewer and improved the discussion due to new calculations requested by the other reviewers.
Submission Date
29 December 2018
Date of this review
13 Jan 2019 20:44:44
Reviewer 2 Report
This is an excellent paper on a very large data set from the Boston Marathon. I only have 1 major concern related to the winning and faster (top 10 or 100)times. Times came down by steadily until the later 70s and have been remarkably stable since then with a few exceptions.
https://en.wikipedia.org/wiki/List_of_winners_of_the_Boston_Marathon
So how do the slower wining and top 10 times prior to mass participation influence your results? Also more recently my guess is that the top 100 times may be getting slower as the sub-elite field has thinned.
Author Response
Reviewer 2
Open Review
(x) I would not like to sign my review report
( ) I would like to sign my review report
English language and style
( ) Extensive editing of English language and style required
( ) Moderate English changes required
(x) English language and style are fine/minor spell check required
( ) I don't feel qualified to judge about the English language and style
Yes | Can be improved | Must be improved | Not applicable | |
Does the introduction provide sufficient background and include all relevant references? | (x) | ( ) | ( ) | ( ) |
Is the research design appropriate? | (x) | ( ) | ( ) | ( ) |
Are the methods adequately described? | (x) | ( ) | ( ) | ( ) |
Are the results clearly presented? | (x) | ( ) | ( ) | ( ) |
Are the conclusions supported by the results? | (x) | ( ) | ( ) | ( ) |
Comments and Suggestions for Authors
This is an excellent paper on a very large data set from the Boston Marathon. I only have 1 major concern related to the winning and faster (top 10 or 100)times. Times came down by steadily until the later 70s and have been remarkably stable since then with a few exceptions.
https://en.wikipedia.org/wiki/List_of_winners_of_the_Boston_Marathon
So how do the slower wining and top 10 times prior to mass participation influence your results? Also more recently my guess is that the top 100 times may be getting slower as the sub-elite field has thinned.
Answer: We agree with the expert reviewer that changes in participation rates across calendar years influenced the race times and consequently the findings. That’s why we used multi-variable statistical models examining both the effects of weather conditions and calendar years on race time (see results section and new Figure 4). We mentioned this aspect in the limitations/strength section before conclusions (“Moreover, it should be highlighted that the applied multi-variable statistical models considered the effects of calendar years and weather conditions on race time. This statistical approach allowed partitioning out the variation of race time across calendar years, since it has been well documented that the increase number of finishers resulted in overall slower marathon race times”). Also, we added in this section that “Also, it was acknowledged that the classification of marathon runners into performance groups such as top10 and top100 referred to different relative numbers of runners (as percentage of the total number) across calendar years. For instance, top 10 was corresponding to a gradually decreased percentage of runners (i.e., became more “elite”) as the number of finishers increased across calendar years. However, these performance groups were selected for the present analysis due to their practical relevance for marathon and their use in the existing literature.”
Submission Date
29 December 2018
Date of this review
11 Jan 2019 20:43:52
Reviewer 3 Report
In the contrast into the Comrades Marathon, the topography of the Boston Marathon is downhill overall, the exception being the Newton Hills at 17 miles.
There is often a predominant westerly wound at the runners the backs until arriving in Boston at 20 miles when there may be an offshore breeze to the finish line.
The race has never been qualified as "official" in terms of marathon records for the reasons above and its unidirectional course rather than being point–2–point.
Author Response
Reviewer 3
Open Review
(x) I would not like to sign my review report
( ) I would like to sign my review report
English language and style
( ) Extensive editing of English language and style required
( ) Moderate English changes required
(x) English language and style are fine/minor spell check required
( ) I don't feel qualified to judge about the English language and style
Yes | Can be improved | Must be improved | Not applicable | |
Does the introduction provide sufficient background and include all relevant references? | (x) | ( ) | ( ) | ( ) |
Is the research design appropriate? | (x) | ( ) | ( ) | ( ) |
Are the methods adequately described? | (x) | ( ) | ( ) | ( ) |
Are the results clearly presented? | (x) | ( ) | ( ) | ( ) |
Are the conclusions supported by the results? | (x) | ( ) | ( ) | ( ) |
Comments and Suggestions for Authors
In the contrast into the Comrades Marathon, the topography of the Boston Marathon is downhill overall, the exception being the Newton Hills at 17 miles.
There is often a predominant westerly wound at the runners the backs until arriving in Boston at 20 miles when there may be an offshore breeze to the finish line.
The race has never been qualified as "official" in terms of marathon records for the reasons above and its unidirectional course rather than being point–2–point.
Answer: We thank the expert reviewer for his/her comments, no specific changes are required.
Submission Date
29 December 2018
Date of this review
11 Jan 2019 00:39:54
Reviewer 4 Report
This study investigated effects of weather conditions on male performance in ‘Boston 16 Marathon’ from 1897 to 2018.
To improve the manuscript, the authors should include the following aspects: - Hypothesis of the study. - Include the registration number of the approval of the Ethics Committee. - Include future perspectives of the study - Indicate the practical repercussion of the results obtainedAuthor Response
Reviewer 4
Open Review
(x) I would not like to sign my review report
( ) I would like to sign my review report
English language and style
( ) Extensive editing of English language and style required
( ) Moderate English changes required
(x) English language and style are fine/minor spell check required
( ) I don't feel qualified to judge about the English language and style
Yes | Can be improved | Must be improved | Not applicable | |
Does the introduction provide sufficient background and include all relevant references? | ( ) | (x) | ( ) | ( ) |
Is the research design appropriate? | (x) | ( ) | ( ) | ( ) |
Are the methods adequately described? | (x) | ( ) | ( ) | ( ) |
Are the results clearly presented? | (x) | ( ) | ( ) | ( ) |
Are the conclusions supported by the results? | (x) | ( ) | ( ) | ( ) |
Comments and Suggestions for Authors
This study investigated effects of weather conditions on male performance in ‘Boston 16 Marathon’ from 1897 to 2018.
To improve the manuscript, the authors should include the following aspects: - Hypothesis of the study. - Include the registration number of the approval of the Ethics Committee. - Include future perspectives of the study - Indicate the practical repercussion of the results obtained
Answer: We thank the expert reviewer and added our hypothesis by inserting ^’We hypothesized to find difference regarding existing literature when investigating the whole period of time (i.e. 1897 to 2018) and including all male annual finishers as well as the annual top 10 and the annual top 100 male finishers’. Regarding the ethical approval, we mention ‘All procedures used in the study were approved by the Institutional Review Board of Kanton St. Gallen, Switzerland, with a waiver of the requirement for informed consent of the participants given the fact that the study involved the analysis of publicly available data (01/06/2010)’. Regarding future perspectives, we added in the end of the discussion ‘Future studies need to confirm these findings for other large city marathons such as the Berlin Marathon or the New York City Marathon’. For the practical applications, we inserted ‘For athletes and coaches, the selection of a marathon race with optimal environmental conditions (e.g. low temperature, low wind, no precipitations) might be important to achieve a fast marathon race time’ at the end of the discussion.
Submission Date
29 December 2018
Date of this review
11 Jan 2019 11:29:33
Reviewer 5 Report
Thank you for the opportunity to review this manuscript submitted to the International Journal of Environmental Research and Public Health. The researchers conducted an analysis of data on male Boston marathon finishers from 1897 to 2018. The purpose of the analysis was to examine the associations between weather conditions and performance among all finishers as well more elite runners. I do think this sort of analysis has value, but unfortunately, I have major concerns with the authors’ overall approach, analysis, and interpretations. Below are my major concerns with the manuscript.
The authors need to be much more transparent about how the weather information was obtained and verified. Why were 4 different websites used? Did each of these websites obtain the weather data from the same weather stations? Also, with any race you’d expect temperature to gradually rise throughout the day, and wind speeds and even wind directions to change. How was all of that information collated? What did the authors do if the websites reported different weather conditions for a given year? The authors literally only discuss the sourcing of weather data in one sentence despite the fact that it’s the most important data in their paper.
To me, it seems likely that the data for temperature, humidity, pressure, and precipitation are all inter-related and are potential confounders of one another. I understand that the authors used a multivariate model to try to address whether each variable had its own independent effects, but I suspect that the analysis may not have dealt with all the issues of multicolinearity.
Are the authors confident that the associations between all the weather-related variables and performance are linear? For example, isn’t it plausible that, among the entire sample, temperature would show a U-shaped relationship with performance? Based on the data presented in Table 1, that seems like a real possibility. The same may also be true for wind speed.
I think including the nationality/geographical region of the runners as a variable over-complicates the analysis. At a minimum, I’m not sure that the regression analysis the authors chose actually addresses what they were trying to achieve. To me, it seems like they are hypothesizing that there’s an interaction between nationality and weather-related variables on performance, or some sort of effect moderation. That doesn’t appear to be what the analysis tested, however.
The paper is lacking when it comes to proposed mechanistic rationales. Why do the authors think there would be an association between barometric pressure and performance? What about for precipitation and performance? While it’s fairly obvious why temperature and wind direction/speed would be associated with performance, it’s not clear to me why barometric pressure and precipitation would be. If there’s not some sort of proposed mechanism to explain these relationships, the researchers shouldn’t be including them in a predictive model.
The overall presentation of the data in the tables and figures is unclear and not justified in many instances. In addition, manuscript contains a substantial number of grammatical errors.
Is there really a good reason to carry out analyses on the winners as a separate group? The general trends and results appear almost identical as to what is presented for top 10 finishers. The additional data for winners just clutters the paper without adding any real scientific value.
In addition to the abovementioned major concerns, I have some additional comments and suggestions for the authors. I hope they find my suggestions to be useful.
ABSRACT:
--pg 1, lines 23-24: The terms ‘greater’ and ‘lower’ have no scientific meaning. Delete them and rephrase. For example, you could just simply state effect size without any additional descriptive wording.
--pg 1, lines 25-27: When you say that performance worsened by 17 sec with increasing pressure, what magnitude of increase in pressure are you referencing? The same issue is apparent when the authors discuss increases in humidity in the next sentence.
--pg 1, lines 27-30: Why are there no effect sizes or p-values presented for these wind and precipitation comparisons?
INTRODUCTION
--pgs 1-2, lines 40-45: I don’t see the point in discussing the health consequences of heat strain when this study focused exclusively on performance times.
--pg 2, lines 57-64: I can connect the dots as to why the authors discuss surface area to body mass ratios, body fat levels, and ectomorph body types, but they never explicitly explain the relevance of these factors: improved heat dissipation. This paragraph needs to be revised to connect these dots and tie them to weather conditions.
--pg 2, line 67: 36 years doesn’t seem like a short period to me.
--pg 2, lines 74-75: I agree that it may be important to study these other weather-related variables, but the authors should provide some sort of mechanistic rationale as to why they think wind direction/speed and barometric pressure would impact performance. For example, headwinds (especially at a high velocity) may increase the energy cost of running. However, they may also increase convective cooling. This is just one example. I’m also quite curious as to why the authors think barometric pressure would impact performance.
METHODS
--pg 3, lines 110-113: The authors need to be much more transparent about how weather information was obtained and verified.
--pg 3, lines 125-127: Using the term country for a variable name doesn’t make sense if some of the regions included aren’t actually countries (Africa, Europe, etc.). I would use a phrase like geographical region instead. Also, what’s the rationale for grouping Central-South Americans together but not grouping those from Canada and USA?
RESULTS
--pg 4, line 157: Delete the sentence, “Therefore, we had many observations per runner.” It’s obvious that some runners competed on several occasions.
--pg: 4, lines 158-166: Were these comparisons actually done with statistical analyses or are they simply descriptive statistics? If the authors didn’t run any actual statistics on the data for this section of the results, then they should not be making claims about performance being lower or higher under various weather conditions. Instead, they should just ask the reader to refer to Table 1 for the descriptive statistics without making any claims about differences.
--pg 4, lines 159-161: The following statement seems inaccurate. “In fact, the average time was fastest (03:11:14 ± 00:11:11 h:min:sec) compared with performances when temperature was lower or higher.” How can the temperature be higher than 30 degrees Celsius?
--pg 4, lines 168-170: The results for temperature described in this section seem counter to what the authors claim in their prior paragraph. In the prior paragraph, they state that performances were best when the temperature was between 24-30 degrees. How, then, can they find that higher temperatures lead to performance with this analysis? It’s likely because the sample size for the 24-30 degree group was tiny and that they didn’t run any statistics for the results presented in Table 1.
--page 4, line 184: The authors reference figure 1a-1b, but when you look at figure 1 on page 9, there are no letters indicating panels a or b.
--pg 5, Table 1: This table is extremely cluttered and quite difficult to digest as a reader. Also, there seems to be a major issue with the way that the authors categorized temperature. For the 24-30 degree group, the sample size is only N = 20. The authors even claim on page 4 (lines 158-159) that average time was fastest for this temperature range. How can they make this claim based on 20 observations out of more than 380,000? Also, how is even possible that only 20 finishers ran in a Boston marathon race that was between 24-30 degrees Celcius?
--pg 6, Table 2: This table is very difficult to interpret and isn’t presented well. For example, what do the S(YEAR)1, S(YEAR)2, etc. represent?
--pg 7, lines 194-203: Again, were these comparisons actually done with statistical analyses or are they simply descriptive statistics? If the authors didn’t run any actual statistics on the data for this section of the results, then they should not be making claims about performance being lower or higher under various weather conditions.
--pg 8, line 216: The authors reference figure 2a-2b, but when you look at figure 2 on page 10, there are no letters indicating panels a or b. Figures 3 and 4 also have the same problem.
DISCUSSION
--pg 14, lines 308-310: The following statement doesn’t seem to have anything to do with humidity, “A similar finding was previously reported in marathon runners, where the performance of slower runners were more affected by unfavourable weather conditions than their faster counterparts.” How does this statement relate to increasing humidity’s correlation with faster running times?
--pg 14, lines 321-324: The authors should discuss why wind coming from the west would be associated with better performance (i.e., the course starts in Hopkinton, which is east of Boston proper).
--pg 14, lines 322-324: What does the following statement have to do with wind direction or even wind speed? “Particularly, it was shown, compared to exercise in treadmill, the air resistance in running outdoors induced an extra cost of oxygen uptake, which increased as the square of wind velocity.”
Author Response
Reviewer 5
Open Review
(x) I would not like to sign my review report
( ) I would like to sign my review report
English language and style
( ) Extensive editing of English language and style required
(x) Moderate English changes required
( ) English language and style are fine/minor spell check required
( ) I don't feel qualified to judge about the English language and style
Yes | Can be improved | Must be improved | Not applicable | |
Does the introduction provide sufficient background and include all relevant references? | ( ) | ( ) | (x) | ( ) |
Is the research design appropriate? | ( ) | (x) | ( ) | ( ) |
Are the methods adequately described? | ( ) | ( ) | (x) | ( ) |
Are the results clearly presented? | ( ) | ( ) | (x) | ( ) |
Are the conclusions supported by the results? | ( ) | (x) | ( ) | ( ) |
Comments and Suggestion for Authors
Thank you for the opportunity to review this manuscript submitted to the International Journal of Environmental Research and Public Health. The researchers conducted an analysis of data on male Boston marathon finishers from 1897 to 2018. The purpose of the analysis was to examine the associations between weather conditions and performance among all finishers as well more elite runners. I do think this sort of analysis has value, but unfortunately, I have major concerns with the authors’ overall approach, analysis, and interpretations. Below are my major concerns with the manuscript.
The authors need to be much more transparent about how the weather information was obtained and verified. Why were 4 different websites used? Did each of these websites obtain the weather data from the same weather stations? Also, with any race you’d expect temperature to gradually rise throughout the day, and wind speeds and even wind directions to change. How was all of that information collated? What did the authors do if the websites reported different weather conditions for a given year? The authors literally only discuss the sourcing of weather data in one sentence despite the fact that it’s the most important data in their paper.
Answer: We thank the expert reviewer and specified in the method section ‘wunderground.com/history/airport/KBOS/2013/1/15/MonthlyHistory.html for 1930 to 2018 and from http://w2.weather.gov/climate/local_data.php?wfo=box for 1897 to 1929’. The problem was that the time frame from 1897 to 2018 was not supported by one weather data archive. A further problem was that only a few data were recorded in earlier years while nowadays plenty of weather data are recorded. We also added in the limitations ‘A limitation is also that we used daily but not hourly weather data from the start of the race until the end of the race. This might have had an influence on the outcome of the results’.
To me, it seems likely that the data for temperature, humidity, pressure, and precipitation are all inter-related and are potential confounders of one another. I understand that the authors used a multivariate model to try to address whether each variable had its own independent effects, but I suspect that the analysis may not have dealt with all the issues of multicollinearity.
Answer: We thank the expert reviewer for this comment. The reviewer is right. We have added a Figure with the pairwise correlation coefficients between weather variables. The highest correlation coefficient related to the pair temperature-humidity and was of -0.45. Therefore, weather variables were not so highly correlated. Anyway, we agree with the reviewer and we have checked the VIF (variance inflation factor) score for each predictor and retained, in the final multi-variable models, only explanatory variable with VIF not greater than 10.
Are the authors confident that the associations between all the weather-related variables and performance are linear? For example, isn’t it plausible that, among the entire sample, temperature would show a U-shaped relationship with performance? Based on the data presented in Table 1, which seems like a real possibility. The same may also be true for wind speed.
Answer: We thank the expert reviewer for this comment. The reviewer is right about the possibility of trend lines, different from the linear ones, to better model the relationship between performance and weather variables. We helped to visualize this curves in Figures 1-3. However, in the final multi-variable models, we consider only the linear hypothesis because we checked that introducing a quadratic term for the temperature and wind speed would have caused multi-collinearity problems. This could be solved using the orthogonal-polynomial fitting but doing this, the model would have been difficult to interpret, since the coefficients would not have been given in terms of the predictor but in terms of a transformation of the predictor.
I think including the nationality/geographical region of the runners as a variable over-complicates the analysis. At a minimum, I’m not sure that the regression analysis the authors chose actually addresses what they were trying to achieve. To me, it seems like they are hypothesizing that there’s an interaction between nationality and weather-related variables on performance, or some sort of effect moderation. That doesn’t appear to be what the analysis tested, however.
Answer: We thank the reviewer for this comment and we agree with him. We have dropped the geographical region variable from the analysis. Now we focused only on the weather conditions.
The paper is lacking when it comes to proposed mechanistic rationales. Why do the authors think there would be an association between barometric pressure and performance? What about for precipitation and performance? While it’s fairly obvious why temperature and wind direction/speed would be associated with performance, it’s not clear to me why barometric pressure and precipitation would be. If there’s not some sort of proposed mechanism to explain these relationships, the researchers shouldn’t be including them in a predictive model.
Answer: We thank the reviewer for this comment. We added more details on these aspects in both introduction and discussion.
The overall presentation of the data in the tables and figures is unclear and not justified in many instances. In addition, manuscript contains a substantial number of grammatical errors.
Answer: We thank the reviewer for having noticed that. We have improved the presentation of the data and results and did our best to check for grammatical errors.
Is there really a good reason to carry out analyses on the winners as a separate group? The general trends and results appear almost identical as to what is presented for top 10 finishers. The additional data for winners just clutters the paper without adding any real scientific value.
Answer: We agree with the expert reviewer and we have eliminated the analysis of the winners.
In addition to the abovementioned major concerns, I have some additional comments and suggestions for the authors. I hope they find my suggestions to be useful.
Answer: We thank the reviewer for these useful suggestions.
ABSTRACT:
--pg 1, lines 23-24: The terms ‘greater’ and ‘lower’ have no scientific meaning. Delete them and rephrase. For example, you could just simply state effect size without any additional descriptive wording.
Answer: We thank the reviewer for this comment. We have deleted them and rephrased.
--pg 1, lines 25-27: When you say that performance worsened by 17 sec with increasing pressure, what magnitude of increase in pressure are you referencing? The same issue is apparent when the authors discuss increases in humidity in the next sentence.
Answer: We thank the reviewer for this suggestion. We have changed the sentence.
--pg 1, lines 27-30: Why are there no effect sizes or p-values presented for these wind and precipitation comparisons?
Answer: We thank the reviewer for this suggestion. We have added effect sizes and p-values relative to the precipitations but not to the wind comparison, otherwise the abstract would have been too long.
INTRODUCTION
--pgs 1-2, lines 40-45: I don’t see the point in discussing the health consequences of heat strain when this study focused exclusively on performance times.
Answer: We agree with the Expert Reviewer and deleted this phrase (“A combination of high temperature and humidity might deteriorate health of marathon runners, e.g. 25% of runners had an acute medical complain in Pittsburgh marathon 1986 (temperature 30°C, humidity 64%) [2].”).
--pg 2, lines 57-64: I can connect the dots as to why the authors discuss surface area to body mass ratios, body fat levels, and ectomorph body types, but they never explicitly explain the relevance of these factors: improved heat dissipation. This paragraph needs to be revised to connect these dots and tie them to weather conditions.
Answer: We agree with the expert reviewer and revised the first paragraph adding “considering their role on heat dissipation” in line 43-44 and “In this context, the relationship between environmental conditions and marathon race time would offer a framework to study the thermoregulatory ability of humans during endurance exercise.” in the end of the paragraph.
--pg 2, line 67: 36 years doesn’t seem like a short period to me.
Answer: We investigate here a period of 121 years, so 36 years is about one third and therefore a rather short period of time
--pg 2, lines 74-75: I agree that it may be important to study these other weather-related variables, but the authors should provide some sort of mechanistic rationale as to why they think wind direction/speed and barometric pressure would impact performance. For example, headwinds (especially at a high velocity) may increase the energy cost of running. However, they may also increase convective cooling. This is just one example. I’m also quite curious as to why the authors think barometric pressure would impact performance.
Answer: We agree with the expert reviewer and developed the rationale for the study before the aims (“Precipitation might influence race time since the occurrence of rain has been shown to moderately inversely correlated with variation in race time [5]. Moreover, knowledge of the effect of wind speed and direction on race time would enhance our understanding of whether a high speed wind increased (increased the energy cost of running depending on the wind direction) or decreased race time or whether an absence of wind increased the race time possibly due to a decrease in convective heat loss [18]. With regards to atmospheric pressure, it has been suggested that an elevated pressure might increase the inspired oxygen [19], whereas might be related the development of precipitation [20].”).
METHODS
--pg 3, lines 110-113: The authors need to be much more transparent about how weather information was obtained and verified.
Answer: We explain now in the method section by changing to ‘Weather data were obtained from two different websites for this period from 1897 to 2018 from www.wunderground.com/history/airport/KBOS/2013/1/15/MonthlyHistory.html for 1930 to 2018 and from http://w2.weather.gov/climate/local_data.php?wfo=box for 1897 to 1929.
--pg 3, lines 125-127: Using the term country for a variable name doesn’t make sense if some of the regions included aren’t actually countries (Africa, Europe, etc.). I would use a phrase like geographical region instead. Also, what’s the rationale for grouping Central-South Americans together but not grouping those from Canada and USA?
Answer: We thank the reviewer for this comment. However this is not anymore applicable, since we have deleted all analysis about country.
RESULTS
--pg 4, line 157: Delete the sentence, “Therefore, we had many observations per runner.” It’s obvious that some runners competed on several occasions.
Answer: We thank the reviewer for having noticed that. We have deleted the sentence.
--pg: 4, lines 158-166: Were these comparisons actually done with statistical analyses or are they simply descriptive statistics? If the authors didn’t run any actual statistics on the data for this section of the results, then they should not be making claims about performance being lower or higher under various weather conditions. Instead, they should just ask the reader to refer to Table 1 for the descriptive statistics without making any claims about differences.
Answer: We thank the reviewer for this comment. We did statistical analysis such as t-test/ANOVA and Dunn’s test for multiple comparisons. We added this in the method section and mentioned it in the results part when differences were not significant. Anyway, we did not report the relative p-values in the table for two reasons. First, the table would have been too large. Second, because we did not want to focus on inference about summary statistics. They were useful to give an overview of the weather effects on performance, but summary statistics did not take into account repeated measurements, for instance. Therefore, these effects had been analyzed rigorously through the multivariable analysis (and uni-variable analysis for pressure and humidity since they had been dropped from the multi-variable models for multi-collinearity reasons).
--pg 4, lines 159-161: The following statement seems inaccurate. “In fact, the average time was fastest (03:11:14 ± 00:11:11 h:min:sec) compared with performances when temperature was lower or higher.” How can the temperature be higher than 30 degrees Celsius?
Answer: We thank the reviewer for this comment. We have changed the sentence.
--pg 4, lines 168-170: The results for temperature described in this section seem counter to what the authors claim in their prior paragraph. In the prior paragraph, they state that performances were best when the temperature was between 24-30 degrees. How, then, can they find that higher temperatures lead to performance with this analysis? It’s likely because the sample size for the 24-30 degree group was tiny and that they didn’t run any statistics for the results presented in Table 1.
Answer: We agree with the reviewer we have explained that, however, this value was the mean of only 20 observations, the only ones recorded in 1909, where participation was low and the average temperature, 28 °C, was the highest, compared to the temperature observed in other years. This is a bias due to the limitation of the dataset already stated in our previous research [Knechtle, B.; Di Gangi, S.; Rust, C.A.; Rosemann, T.; Nikolaidis, P.T. Men's participation and performance in the Boston marathon from 1897 to 2017. International journal of sports medicine 2018.].
--page 4, line 184: The authors reference figure 1a-1b, but when you look at figure 1 on page 9, there are no letters indicating panels a or b.
Answer: We thank the reviewer for this comment. However, the Figure has been removed.
--pg 5, Table 1: This table is extremely cluttered and quite difficult to digest as a reader. Also, there seems to be a major issue with the way that the authors categorized temperature. For the 24-30 degree group, the sample size is only N = 20. The authors even claim on page 4 (lines 158-159) that average time was fastest for this temperature range. How can they make this claim based on 20 observations out of more than 380,000? Also, how is even possible that only 20 finishers ran in a Boston marathon race that was between 24-30 degrees Celcius?
Answer: We thank the reviewer for having noticed that. We hope that the table is more readable after eliminating the winner category. For the strange results about temperature 24-30 see the answer to the previous comment about pg 4, lines 168-170.
--pg 6, Table 2: This table is very difficult to interpret and isn’t presented well. For example, what do the S(YEAR)1, S(YEAR)2, etc. represent?
Answer: We thank the reviewer for this comment. We have changed it and clarified the definition of S(YEAR) terms.
--pg 7, lines 194-203: Again, were these comparisons actually done with statistical analyses or are they simply descriptive statistics? If the authors didn’t run any actual statistics on the data for this section of the results, then they should not be making claims about performance being lower or higher under various weather conditions.
Answer: We thank the reviewer for this comment. The question had been already answered (comment about pg: 4, lines 158-166).
--pg 8, line 216: The authors reference figure 2a-2b, but when you look at figure 2 on page 10, there are no letters indicating panels a or b. Figures 3 and 4 also have the same problem.
Answer: We thank the reviewer for this comment. However, the Figure has been removed.
DISCUSSION
--pg 14, lines 308-310: The following statement doesn’t seem to have anything to do with humidity, “A similar finding was previously reported in marathon runners, where the performance of slower runners were more affected by unfavourable weather conditions than their faster counterparts.” How does this statement relate to increasing humidity’s correlation with faster running times?
Answer: We agree with the expert reviewer and deleted that sentence.
--pg 14, lines 321-324: The authors should discuss why wind coming from the west would be associated with better performance (i.e., the course starts in Hopkinton, which is east of Boston proper).
Answer: We agree with the expert reviewer and specified to ‘The race starts in Hopkinton, located to the west-south-west of Boston. The athletes run in the general direction west-south-west and wind coming from west, compared to wind from other directions, was the most favourable for performance of all groups of finishers’.
--pg 14, lines 322-324: What does the following statement have to do with wind direction or even wind speed? “Particularly, it was shown, compared to exercise in treadmill, the air resistance in running outdoors induced an extra cost of oxygen uptake, which increased as the square of wind velocity.”
Answer: We agree with the expert reviewer and deleted that sentence.
Submission Date
29 December 2018
Date of this review
04 Jan 2019 21:38:57
Round 2
Reviewer 1 Report
Congratulations for tour work
Author Response
as far as we are aware no further changes are required from reviewer 1